# Spatial and Temporal Pattern of Net Ecosystem Productivity in China and Its Response to Climate Change in the Past 40 Years

**DOI:** 10.3390/ijerph20010092

**Published:** 2022-12-21

**Authors:** Cuili Zhang, Ni Huang, Li Wang, Wanjuan Song, Yuelin Zhang, Zheng Niu

**Affiliations:** 1State Key Laboratory of Remote Sensing Science, Aerospace Information Research Institute, Chinese Academy of Sciences, Beijing 100094, China; 2University of Chinese Academy of Sciences, Beijing 100049, China

**Keywords:** China, net ecosystem productivity, spatiotemporal changes, trend analysis, partial correlation analysis

## Abstract

Net ecosystem productivity (NEP), which is considered an important indicator to measure the carbon source/sink size of ecosystems on a regional scale, has been widely studied in recent years. Since China's terrestrial NEP plays an important role in the global carbon cycle, it is of great significance to systematically examine its spatiotemporal pattern and driving factors. Based on China's terrestrial NEP products estimated by a data-driven model from 1981 to 2018, the spatial and temporal pattern of China's terrestrial NEP was analyzed, as well as its response to climate change. The results demonstrate that the NEP in China has shown a pattern of high value in the west and low value in the east over the past 40 years. NEP in China from 1981 to 2018 showed a significantly increasing trend, and the NEP change trend was quite different in two sub-periods (i.e., 1981–1999 and 2000–2018). The temporal and spatial changes of China's terrestrial NEP in the past 40 years were affected by both temperature and precipitation. However, the area affected by precipitation was larger. Our results provide a valuable reference for the carbon sequestration capacity of China's terrestrial ecosystem.

## 1. Introduction

Global warming has caused sea level rise, floods, droughts, pests, and extreme weather events (e.g., La Nina and El Nino), which directly affect the long-term evolution of the global carbon cycle [1,2]. In recent years, many countries have taken a series of response measures to effectively reduce greenhouse gas emissions, improve the Earth's environment, and maintain the sustainable development of the global social economy [3,4]. China has also set the emission reduction target of “striving to reach the peak of the carbon dioxide emissions by 2030 and achieve carbon neutrality by 2060”. An accurate assessment of China’s terrestrial carbon cycle process and its feedback on climate change is considered the key to estimating future CO_2_ concentration and predicting climate change effects accurately. It is also a comprehensive and large-scale hot issue [5].

Net ecosystem productivity (NEP) is defined as the difference between the carbon fixed by the photosynthesis of the ecosystem and the carbon lost by the respiration of the ecosystem [6]. It represents the change rate of net carbon flux or carbon stock between land and atmosphere. The concept of NEP is proposed to analyze the carbon sink/source function of the terrestrial biosphere, representing the net photosynthetic yield of atmospheric CO_2_ entering the ecosystem. Without considering the various disturbance effects, NEP quantitatively reflects the size of the terrestrial ecosystem carbon source/sink. However, there are still great uncertainties in the deep understanding of the temporal and spatial changes, as well as the driving factors of NEP in China's terrestrial ecosystem. For example, Tao et al. [7] estimated China's land NEP from 1981 to 2000 by using a high-resolution climate database and ecosystem process model. The authors found that the 20-year average value of NEP was higher in the north and lower in the south, higher in the middle and southwest, and lower in the southeast. In another interesting work, Cao et al. [8] projected China’s land NEP from 1981 to 2000 by using a process-based biogeochemical model and a remote sensing-based production efficiency model. The authors found that China's terrestrial ecosystems were taking up carbon, but the capacity was significantly undermined by ongoing climate change. Yu et al. [9] quantitatively estimated the NEP of the different terrestrial ecosystems in China, and found that it presents an obvious latitude pattern, while its spatial change is mainly affected by the annual average temperature. Zhang et al. [10] investigated the contribution of different climate factors to the interannual change of NEP, and revealed that the annual variability of NEP in China is mainly affected by the variability of the precipitation.

China has been constantly improving its ecosystem environment through a series of ecological restoration projects in recent decades, which has greatly changed the terrestrial carbon budget of China [11,12]. Therefore, a further study of the temporal and spatial pattern of the terrestrial NEP in China and its response to climate changes in the past 40 years can not only offer a data basis for understanding the relationship between China's terrestrial ecosystem and climate change, but also provide a solid basis for formulating climate change response measures.

In this study, Theil–Sen median trend analysis and the Mann–Kendall test were used to analyze the temporal change trend of terrestrial NEP in China in the past 40 years. In addition, the partial correlation analysis method was used to analyze the relationships between NEP and climate driving factors (annual average air temperature and annual total precipitation). The spatiotemporal pattern of NEP in China's terrestrial ecosystem from 1981 to 2018 was also revealed and its response to climate change was systematically evaluated.

## 2. Materials and Methods

### 2.1. Data Source and Processing

The NEP data used in this work were from the National Earth System Science Data Center, the national science and technology basic conditions platform (http://www.geodata.cn, accessed on 8 March 2022). This product had a spatial resolution of 5 km and a temporal resolution of one month. The global terrestrial NEP was estimated by using a random forest model and multisource remote sensing data for the global vegetation distribution area, excluding bare land, water areas, construction land, ice, and snow-covered areas and other non-vegetation areas [13]. The climate data in this work were from ERA5 reanalysis data with a spatial resolution of 9 km and temporal resolution of one month (https://www.ecmwf.int/en/forecasts/datasets/reanalysis-datasets/era5, accessed on 25 March 2022). The NEP and the climate data of China from 1981 to 2018 were obtained by using the boundary data of China, while the annual data was obtained by accumulating monthly data. Therefore, our study area covers the vegetation area of China. In addition, the nearest neighbor resampling method was utilized to obtain annual climate data with a spatial resolution of 5 km.

### 2.2. Temporal Change Trend of NEP

A combination of Theil–Sen median trend analysis and the Mann–Kendall test was used to analyze temporal change trends in terrestrial NEP in China. The Theil–Sen estimator is an unbiased estimation of the true slope in the linear regression algorithm without the need to use a normal distribution of data. Theil–Sen estimates are more robust and less susceptible to outlier data than least squares estimates [14,15]. Theil–Sen median trend analysis employs the median of the slopes of all data pairs as an estimate of the overall slope of change in the time series by calculating the slope between the two pairs of data on the time series, which is calculated by the following expression (Equation (1)):(1)$$\overset{⌒}{\beta }=Median\left(\frac{x_{j}-x_{i}}{j-i}\right),\forall 1\leq i\leq j\leq n$$

The Mann–Kendall test [16], as a complement to the Theil–Sen median trend analysis, can determine the significance of the time series trends (Equations (2) and (3)). Therefore, the combination of the Theil–Sen median trend analysis and Mann–Kendall test has been commonly applied for the analysis of remote sensing long time series [17,18].
(2)$$\mathrm{sgn}(x_{j}-x_{i})=\left\{\begin{array}{c} +1,\begin{array}{c} x_{j}-x_{i}>0 \end{array}\\ 0,\begin{array}{c} x_{j}-x_{i}=0 \end{array}\\ -1,\begin{array}{c} x_{j}-x_{i}<0 \end{array} \end{array}\right.$$
(3)$$S=\sum_{i=1}^{n-1}\sum_{j=i+1}^{n}\mathrm{sgn}(x_{j}-x_{i})$$

In Equations (1)–(3), $x_{j}$ and $x_{i}$ are the data of the adjacent points on the time series data, and *n* is the number of time series data. When n≤10, the statistic S can be directly tested for the bilateral trend. At a given significance level α, if $\left|S\right|\geq S_{\alpha /2}$, the original hypothesis can be rejected and the original time series is considered to have a significant trend. Otherwise, the original hypothesis is accepted and the trend in the time series is considered to be insignificant.

When n>10, the statistic S approximately obeys the standard normal distribution, and the standardized statistic S yields the statistic *Z*. The value of *Z* can be calculated by Equation (4).
(4)$$Z_{S}=\left\{\begin{array}{c} \frac{S-1}{\sqrt{Var\left(S\right)}},\begin{array}{cc} if & S>0 \end{array}\\ 0,\begin{array}{cc} \;if & S=0 \end{array}\\ \frac{S-1}{\sqrt{Var\left(S\right)}},\begin{array}{cc} if & S<0 \end{array} \end{array}\right.$$
where
(5)$$Var\left(S\right)=\frac{n\left(n-1\right)\left(2n+5\right)-\sum_{i=1}^{m}t_{i}\left(t_{i}-1\right)\left(2t_{i}+5\right)}{18}$$

In Equations (4) and (5), *n* is the number of time series data, *m* is the number of knots (recurring data sets) in the sequence, and $t_{i}$ is the width of the knot (the number of duplicate data in the *i*th duplicate data set). The *Z* statistic was also tested by the bilateral trend method and the critical value $(Z_{1-\alpha /2}$) was obtained in the normal distribution table at a given significant level α. More specifically, if $\left|Z\right|\leq Z_{1-\alpha /2}$, the original hypothesis is accepted and the trend of the time series is not significant. In the opposite case, where $\left|Z\right|>Z_{1-\alpha /2}$, the original hypothesis is rejected and the trend of the time series data is considered significant.

### 2.3. Response Analysis of NEP to Climate Factors

This study analyzed the response of NEP to climate factors (air temperature, precipitation) by using the partial correlation method [19]. First, the correlation coefficient between the NEP and the air temperature or precipitation was calculated by using Equation (6).
(6)$$R_{xy}=\frac{\sum_{i=1}^{n}\left[\left(x_{i}-\bar{x}\right)\left(y_{i}-\bar{y}\right)\right]}{\sqrt{\sum_{i=1}^{n}(x_{i}-\bar{x}{)}^{2}\sum_{i=1}^{n}(y_{i}-\bar{y}{)}^{2}}}$$

In Equation (6), $R_{xy}$ refers to the correlation coefficient between *x* and *y*, *i* denotes the number of temporal data, *x* represents the NEP in the year *i* and *y* is the air temperature or precipitation in the year *i*. The partial correlation coefficients of the precipitation-based NEP with air temperature and air-temperature-based NEP with precipitation were then calculated by using the following Equation (7).
(7)$$R_{xy\_Z}=\frac{R_{xy}-R_{xz}R_{yz}}{\sqrt{\left(1-R_{xy}{}^{2}\right)}\left(1-R_{yz}{}^{2}\right)}$$
where $R_{xy\_z}$ is the partial correlation coefficient between *x* and *y* after fixing *z*. The significance test of the partial correlation coefficient was performed by t-test and its statistic was calculated as described in Equation (8).
(8)$$t=\frac{R_{xy\_z}}{\sqrt{1-R_{xy\_z}{}^{2}}}\sqrt{n-2}$$

## 3. Results and Discussion

### 3.1. Spatial Distribution Pattern of Multi-Year Averaged NEP in China

The high-value areas of the annual mean NEP from 1981 to 2018 were widely distributed in the tropical and subtropical regions at low latitudes in China (i.e., Yunnan, Guangdong, Fujian, and Jiangxi Provinces). Furthermore, the low-value areas were concentrated in central and eastern China (i.e., Henan, Hebei, Shandong, and Anhui Provinces), the central area of Northeast China, and the eastern area of the Qinghai–Tibet Plateau (Figure 1). Overall, NEP in the different vegetation types showed large regional differences (Figure 1 and Appendix A). The high value was mainly distributed in the subtropical evergreen broad-leaved forest and tropical monsoonal forest areas in southern China (accounting for 82.75% of the total terrestrial NEP in China), followed by the temperate grassland areas in the eastern area of Northeast China and Inner Mongolia (9.53%), and the cold-temperate coniferous forest areas in the Daxing’anling (6.00%). The low-value areas were also widely distributed in the warm-temperate deciduous broadleaf forest areas (65.95%) in the North China Plain, followed by the Qinghai–Tibet Plateau alpine vegetation area (27.25%), and the temperate desert areas (6.80%) in Northwest China. The spatial distribution pattern of NEP in various vegetation types was consistent with the findings of Liang et al. [20] and Jiang et al. [21].

### 3.2. Change Trend of NEP over the Past 40 Years

The mean value of the total annual terrestrial NEP in China from 1981 to 2018 was 397.02 ± 42.72 × 10^12^ g C yr^−1^ (mean ± standard deviation), which was consistent with the previously estimated terrestrial carbon sinks in China over the past 20 years by using different methods [8,22]. From 1981 to 2018, the annual total terrestrial NEP in China exhibited an overall significant increasing trend, with the lowest value of 318.95 × 10^12^ g C yr^−1^ in 1991 and the highest value of 462.96 × 10^12^ g C yr^−1^ in 2017. The increased rate of the total terrestrial NEP in China from 2000 to 2018 (1.32 × 10^12^ g C yr^−2^) was also slightly smaller than that from 1981 to 1999 (1.47 × 10^12^ g C yr^−2^). This result indicated that the carbon sink of the terrestrial ecosystems in China increased significantly in the past 40 years without considering disturbances. However, the increase in the carbon sink was larger from 1981 to 1999 than that from 2000 to 2018. Our results were also consistent with the findings of previously reported works in the literature (Tao et al. [7], Liang et al. [20], and Chuai et al. [23]), which revealed that the NEP of the Chinese terrestrial ecosystems gradually increased after the 1980s with the implementation of a series of ecological restoration projects.

There were large differences in the change trends of terrestrial NEP in China from 1981 to 1999 and from 2000 to 2018 (Figure 2). For example, from 1981 to 1999, 52.67% of the studied area showed an increasing trend in annual NEP and 47.26% of the studied area showed a decreasing trend. However, from 2000 to 2018, the areas with an increasing and a decreasing trend were 60.42% and 39.52%, respectively. From 1981 to 1999, the areas with significantly increased NEP were mainly located in the southern part of Shanxi Province, the western border of Hubei Province, and the southeastern coastal region, while the areas with significantly decreased NEP were concentrated in the northern part of the Loess Plateau, as well as the central part of Northeast China. From 2000 to 2018, the areas with significantly increased NEP were mainly located in the southern part of the Loess Plateau, the southern part of Henan Province, and the central part of Northeast China, while the significantly decreased NEP was mainly located in the Daxing’anling, Xiaoxing’anling, and Sanjiang Plain areas of Northeast China, and the central part of Hebei Province.

The difference in the changing trends of NEP for the two periods (i.e., 1981–1999 and 2000–2018) could be accounted for by many reasons. Nevertheless, land cover change from human activities may be one of the main reasons [24,25]. For example, Li et al. [26] demonstrated that conversion of paddy and dryland crops in parts of the Songnen Plain and the Sanjiang Plain took place between 1998 and 2009, which may be the main reason for the change of NEP in the Northeast Plain region. Zhang et al. [27] reported that multiple afforestation programs such as the Three-North Shelterbelt Development Program and the Nature Forest Conservation Program have accelerated greenness in Northeast China since 1982. Before 2000, the NEP of Loess Plateau showed a significant decreasing trend over 70% of the total area. However, from 2000 to 2018, the area of Loess Plateau with an increasing trend reached 63.11%. This was consistent with the work of Chen et al. [28], which indicated that the implementation of ecological restoration projects, such as afforestation, significantly increased the vegetation carbon sequestration in a farming–pastoral ecotone in northern China, on the Loess Plateau, and in southeastern China. Moreover, Lu et al. [29] pointed out that a vegetation change trend in the North China Plain was caused by the transformation of land use (such as intensive cultivation in Henan Province, ecological protection measures in Hebei Province, etc.).

### 3.3. The Responses of Spatiotemporal Changes in NEP to Climate Change

From 1981 to 2018, NEP was significantly correlated with precipitation in nearly 40% of the regions in China, with about 20% of the regions having a significant positive correlation between NEP and precipitation, mainly in the temperate grassland area (southwest of Northeast China, central Inner Mongolia, northern Xinjiang, and the Loess Plateau region). On the contrary, the area with the most significant correlation between NEP and temperature was less than 4% and distributed sporadically, which indicated that the correlation between NEP and temperature in most regions of China was not significant. From 1981 to 1999, NEP in 31.93% of the studied area was significantly correlated with precipitation, only 5.05% with temperature, and 5.24% with both temperature and precipitation. Moreover, from 2000 to 2018, NEP in 33.91% of the study area was significantly correlated with precipitation, 2.93% with temperature, and 2.28% with both temperature and precipitation. These results indicate that the terrestrial NEP in China is strongly influenced by both temperature and precipitation, but the areas influenced by precipitation are obviously larger (Figure 3). Therefore, precipitation maybe the main influencing factor of the NEP changes in terrestrial ecosystems in China. These results are consistent with the outcomes of the previous literature, which indicated that rainfall variability dominates NEP annual variability in China [10], especially in semiarid and arid regions in northern China [30,31]. The work of Chang et al. [32] showed that rain use efficiency increased in Northwest China from 1982 to 2013, which supports the results of our study to some extent. Moreover, Wang et al. [33] also demonstrated that global warming is not conducive to the global carbon sink but abundant rainfall is quite important for the global carbon cycle.

Although there are few regions with a significant correlation between NEP and temperature, the area of NEP correlated with temperature was significantly larger for 1981–1999 than that for 2000–2018, especially in the northwestern part of the Loess Plateau (Figure 3b,d). It is noteworthy that from the end of the 20th century to the beginning of the 21st century, the area affected by precipitation in the Loess Plateau region increased from 30.77% to 72.91%, while the area affected by temperature only decreased from 7.89% to 1.06%. Additionally, the area affected by temperature and precipitation simultaneously decreased from 30.79% to 5.52%. According to the work of Sun et al. [34], the implementation of the Grain for Green Program has gradually increased the proportion of vegetation affected by the precipitation on the Loess Plateau and decreased the proportion of vegetation affected by the temperature, which is consistent with the results of this work. The vegetation on China’s Loess Plateau has been reported to be greatly changed especially due to the Grain for Green Program launched in 1999 [35,36]. Thus, the implementation of ecological restoration programs in the Loess Plateau may lead to changes in the response of NEP to climate change.

Although the role of climate changes in the NEP changes of China’s ecosystems is highlighted in this work, various sources of uncertainties do exist. For example, NEP products and climate data in this work can have significant sources of uncertainty, which could result in uncertainties in the spatial pattern of the terrestrial NEP in China and its response to climate change. Future works should be committed to improving the accuracy of NEP estimation using new technology [37,38], and include other important processes such as CO_2_ fertilization, land use change, and human management (e.g., fertilization and irrigation) [39].

## 4. Conclusions

The high-value areas of China‘s terrestrial NEP are mainly distributed in tropical and subtropical forest areas, while the low-value areas are widely distributed in the central and eastern parts of China and the Qinghai–Tibet Plateau alpine vegetation area. In the past 40 years, the total NEP in China has revealed a significantly increased trend, especially in the tropical and subtropical regions. In addition, the trend of increasing NEP on the Loess Plateau due to the application of ecological projects was obvious, indicating that the implementation of China's natural restoration policy has produced promising results. The response of NEP to both temperature and precipitation varied at different time periods, but precipitation was the main influencing factor of the NEP changes in terrestrial ecosystems in China. The increase in the carbon sink due to the increase in NEP can generate great ecological and economic values. Works on NEP and its influencing factors are crucial for the conservation of biodiversity and environmental health.

## Figures and Tables

**Figure 1 ijerph-20-00092-f001:**
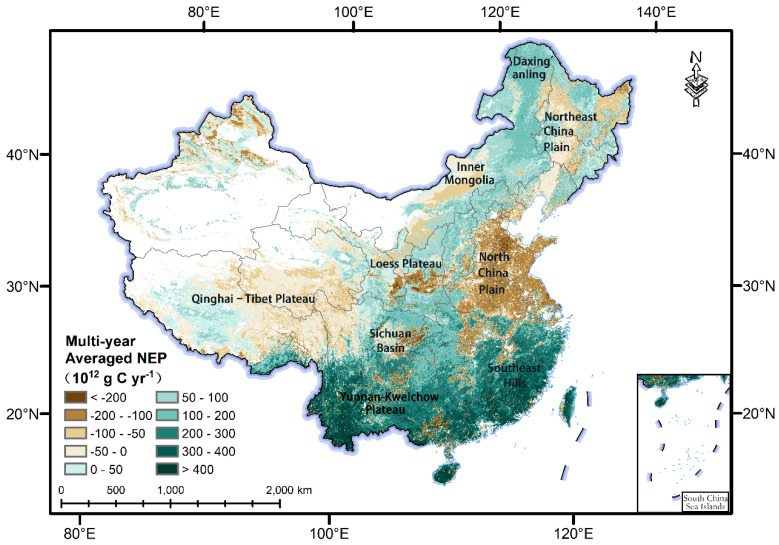
Spatial distribution pattern of the terrestrial multi-year averaged net ecosystem productivity (NEP) in China from 1981 to 2018.

**Figure 2 ijerph-20-00092-f002:**
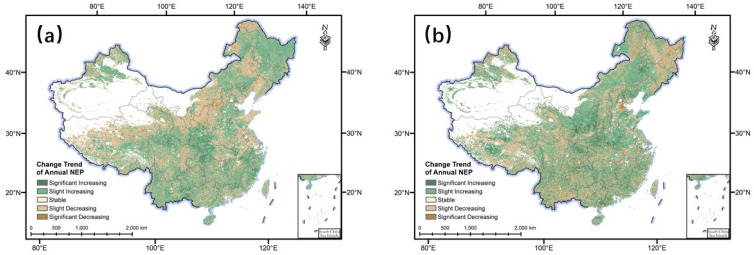
Change trends of the terrestrial net ecosystem productivity (NEP) in China from 1981 to 1999 (**a**) and from 2000 to 2018 (**b**).

**Figure 3 ijerph-20-00092-f003:**
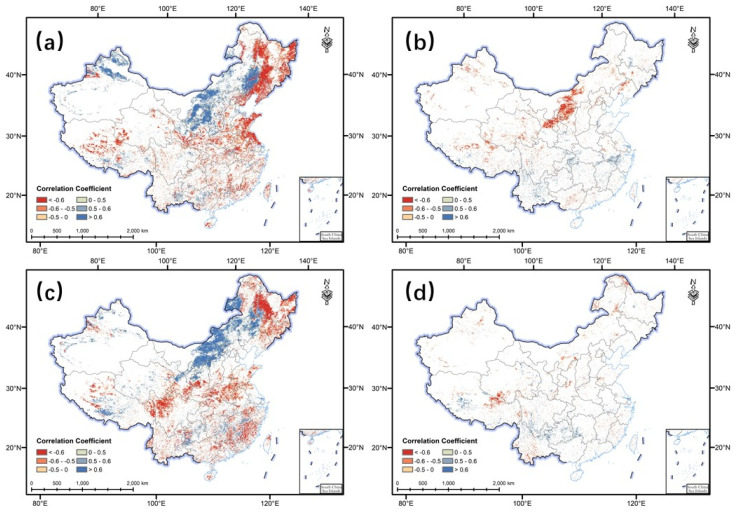
Spatial pattern of the partial correlation coefficients of the terrestrial net ecosystem productivity (NEP) with air temperature and precipitation in China from 1981 to 2018. (**a**) NEP with precipitation from 1981 to 1999; (**b**) NEP with air temperature from 1981 to 1999; (**c**) NEP with precipitation from 2000 to 2018; (**d**) NEP with air temperature from 2000 to 2018.

## Data Availability

The data presented in this study are available on request from the corresponding author.

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
