# Peer review of "Spatial and Temporal Pattern of Net Ecosystem Productivity in China and Its Response to Climate Change in the Past 40 Years"

_ijerph, 2022, doi:10.3390/ijerph20010092_

Round 1

Reviewer 1 Report

    Based on China's terrestrial NEP products estimated by a data-driven model from 1981 to 2018, The authors analyzed the spatial and temporal pattern of China's terrestrial NEP, as well as its response to climate change. The results can offer a data basis for understanding the relationship between China's terrestrial ecosystem and climate change, and provide a solid basis for formulating climate change response measures. 

The following recommendations:

(1) In order to better understand the spatial distribution characteristics of NEP, we suggest that the major region names be labeled in Figure 3.1(Line 139)

(2) In order to show the relationship between eco-environmental planning and vegetation regionalization more clearly, can you add the vegetation map of China? (Line 148)

(3) to better understanding the difference in the changing trends of NEP for the two periods, Can you complement the spatial distribution of ecological projects implemented in China? (Line 191-202)

(4) “High-value areas are mainly distributed in tropical and subtropical forest areas, while low-value areas are widely distributed in the central and eastern parts of China and the alpine vegetation areas of the Tibetan Plateau. In the past 40 years”, (Line 258) --I'm sorry, but I see no direct evidence to support this conclusion.

(5) In the references, some papers giving the  DOI numbers, but some nothing. Please standardize the references.

Author Response

Please see the attachment for a point-by-point response.

Reviewer 2 Report

REVIEW

Journal: International Journal of Environmental Research and Public HealthManuscript ID: ijerph-2053529Type of manuscript: ArticleTitle: Spatial and Temporal Pattern of Net Ecosystem Productivity in Chinaand its Response to Climate Change in Recent 40 Years Authors: Cuili Zhang, Ni Huang *, Li Wang, Yuelin Zhang, Zheng Niu
 This manuscript depicts estimates of Net Ecosystem Productivity (NEP) during 1981-2018 China based on remote sensing data, resolution 5*5km. NEP was observed high in south-Eastern parts of China and low in the west. NEP incresed ovet time, yet the che change from 1981 to 1999 was not identical with the change from the year 2000 to 2018. Temperature and precipitation records were used to anlyse the interannual variation of NEP. The results indicated more significant correlations with precipitation than with temperature.

The methodology is sufficiently clear, the results are interesting and the maps shown (Figs. 3.1.-3.3.) are easy to understand and useful for science communication. The English language is a little inaccurate in places but, nevertheless, acceptable. My main critical remarks refer to the concept NEP and its interpretation. When disturbances such as land clearance and forest harvests have not been included into the analysis, this ”NEP” is not fully proportional to the rate of carbon sequestration (the carbon sink). Perhaps it were more appropriate to use a ”global greening” approach, see e.g.:

Piao, S., Wang, X., Park, T., Chen, C., Lian, X. U., He, Y., ... & Myneni, R. B. (2020). Characteristics, drivers and feedbacks of global greening. Nature Reviews Earth & Environment1(1), 14-27.

I recommend accepting this manuscript for publication after the authors have expanded sections Introduction and Discussion with better explanations of this concept ”NEP” and its relation to other remote sensing indicators such as the ”greenin” and ”browning”.

Author Response

(The authors gave the same response as above.)
